# ROS-induced ATP synthase mRNA degradation and metabolism dysfunction reveals the mechanism of artificial deteriorated cotton seeds

Ci Song, Junming Zhang, Zhenzhen Xing, Fengxu Gu, Junying Chen ⓞ*

Agronomy College of Henan Agricultural University, Zhengzhou, China

* chenjunying3978@126.com

## Abstract

Seed aging is a complex biological process, the deterioration of oil crop seeds, in particular, has caused yield decline and economic loss to a large extent. However, research on the aging mechanism of oilseeds, such as cotton seeds, is still unclear. In this study, the physiological and biochemical changes in artificially aged cotton seeds were examined to further reveal the mechanism of seed aging and deterioration. Cotton seeds of "Xinluzao 74" were treated by artificial aging treatment method with high temperature (45°C) and high relative humidity (100%) for 1, 2, 3, 4, or 5 days, respectively, and untreated cotton seeds were used as control (CK). Our results showed that the germination rate, seed embryo viability, dehydrogenase activity, the activities of dehydrogenase antioxidant enzymes (SOD and POD), seed respiration rate, ATP content, ATP synthase activity and ATP synthase subunit mRNA integrity all showed a significant downward trend with the aging treatment time ($P<0.05$), whereas the ROS generation ($H_2O_2$ content and $\cdot O_2^-$ production rate), relative conductivity, MDA content of seeds increased significantly, and the ultrastructure of cell membrane, mitochondria and chromatin of seed embryo was seriously damaged. Correlation analysis showed that there was a strong negative correlation between germination rate, SOD and POD activities, and respiration rate with $H_2O_2$ content ($P<0.05$). This study reveals that excessive ROS, particularly $H_2O_2$, causes oxidative damage to ATP synthase subunit mRNAs, leading to impaired mitochondrial respiration and reduced seed vigor. These findings provide new molecular evidence linking oxidative RNA damage with seed aging, which could inform seed quality evaluation and storage strategies.

## Introduction

Seed deterioration is a complex biological process that inevitably occurs during storage, yet its molecular basis remains incompletely understood [1]. The primary factors that affect seed deterioration include environmental temperature, humidity,

**Data availability statement:** All relevant data are within the paper and its Supporting Information files.

**Funding:** This work was supported by the National Science Foundation of China (NSFC Grant No. 31571761, and No. 31971998) and National Key Research and Development Program of China (No. 2018YFD0101003-02). The funders had no role in the study design, data collection and analysis, decision to publish, or preparation of the manuscript.

**Competing interests:** The authors have declared that no competing interests exist.

physiological state of the seed and its chemical composition. Seeds of oil crops are particularly susceptible to oxidative deterioration due to their high levels of unsaturated fatty acids [2–5]. As major sources of edible, pharmaceutical, and industrial oils, oilseeds—such as soybeans, peanuts, and cotton seeds—play a vital role in global nutrition and economy. Deterioration of oilseeds not only reduces germination ability and crop yield, but also threatens crop diversity, global industrial and economic development. Therefore, understanding the molecular mechanisms underlying oilseed deterioration is essential for ensuring food security and promoting sustainable agriculture.

Reactive oxygen species (ROS)-induced oxidative damage has been identified as a key factor in seed deterioration [6,7]. Non-enzymatic reactions, including the Amadori and Maillard reactions [8], occur spontaneously in cells during dry storage of seeds and can promote ROS accumulation. This can disrupt the cellular antioxidant system, particularly affecting key enzymes such as peroxidase (POD), superoxide dismutase (SOD), and glutathione reductase (GR), which are highly sensitive to aging [9]. Oxidative damage in seeds, caused by reduced antioxidant enzyme activity, ultimately leads to decreased seed viability [6,10]. During seed imbibition, a number of enzymatic reactions, such as mitochondrial respiration, can increase ROS generation and induce oxidative damage. For example, superoxide anions—a toxic byproduct of mitochondrial respiration—can adversely affect cell viability [11]. ROS can migrate across the plasma membrane freely, and attack any possible subcellular site (e.g., mitochondria, chromatin), causing irreversible damage to organelle structure and function [12,13]. ROS also react with macromolecules, triggering lipid peroxidation, enzyme inactivation, and nucleic acid degradation [7,14], ultimately reducing seed vigor and germination capacity. However, although oxidative damage to DNA and lipids has been widely studied, oxidation of mRNA and its effects on mitochondrial metabolism during seed aging remain largely unknown.

Mitochondria serve as central hubs for energy synthesis and metabolism regulation [15,16]. The substances and energy required during seed germination also depend on the initiation of metabolic pathways including glycolysis, the tricarboxylic acid (TCA) cycle, and oxidative phosphorylation [17]. However, mitochondria are particularly vulnerable to free radical attacks than other organelles due to their high membrane surface area-to-volume ratio and high-water content, and role as a major site of ROS generation via oxidative phosphorylation [18,19]. In aged maize seeds, mitochondrial deterioration has been correlated with the loss of viability—mitochondria in aged embryos appeared markedly swollen, and showed distortions of the internal structures (outer membrane and/or cristae) disorganized or almost absent [13,20,21]. Similarly, artificial aging treatments in elm seeds have been shown to induce ROS-mediated mitochondrial membrane damage and loss of mitochondrial transmembrane potential, resulting in an irreversible loss of seed viability [22]. Moreover, oxidative inactivation of metabolic enzymes (e.g., ATP synthase, malate dehydrogenase) may further inhibit ATP generation and impairs energy production [19,23]. Similarly, aged sugar beet seeds exhibit diminished TCA cycle and oxidative phosphorylation enzyme activity, leading to a dysfunction in respiration metabolism

[24]. Respiration gradually weakens over time with the decrease of enzyme activity, resulting in a corresponding reduction in the energy (ATP) production and material supply required [25].

Nucleotides are also susceptible to oxidative damage caused by ROS, malondialdehyde (MDA), and non-enzymatic reactions [6]. ROS can trigger various types of damage, including base modifications, DNA strand breakage, and DNA methylation. Meanwhile, MDA, a product of lipid peroxidation, promotes nucleic acid adducts (e.g., 8-OHdG) and chain fragmentation [6]. Compared to double-stranded DNA, single-stranded RNA is more vulnerable to oxidative damage, and oxidized RNA tends to degrade since it lacks repair mechanisms. A large amount of mRNA is stored in dry seeds, which plays an important role in regulating seed germination [26]. For example, dried seeds of *Arabidopsis thaliana* [27] and rice [28] rely on the mRNA stored in seeds for selective translation to ensure the completion of seed germination. However, mRNA is more vulnerable to oxidative stress than other types of RNA. Oxidation of guanine (G) to 8-dihydroxy guanine (8-OHG) in mRNA can cause premature translation termination and reduced protein synthesis efficiency, thereby impairing seed vigor [23,29].

Cotton (*Gossypium hirsutum L.*) is a crucial global cash crop. Its seeds are rich in unsaturated fatty acids, making them particularly susceptible to oxidative damage during storage, which complicates long-term seed preservation. Previous studies on cotton seed aging focused on physiological indices (e.g., germination rates, SOD/POD activity, MDA levels, leaf conductivity) [30,31]. Our earlier work suggested that cotton seed deterioration may be linked to the dysfunction of the mitochondrial respiratory metabolic system caused by the integrity loss of ATP synthase subunit mRNAs [32]. However, direct evidence of mitochondrial structural damage is still lacking, and the causes of mitochondrial metabolic damage in cotton seeds remain unclear. Therefore, more studies need to be done for elucidating the mechanism of cotton seeds deterioration. "Xinluzao 74" was one of the important cotton varieties cultivated in Xinjiang, China, which has excellent agronomic traits such as high yield, stable growth, disease resistance, and strong adaptability, representing typical seed storage behavior of upland cotton. Its characteristics make it an ideal model for artificial aging studies with global relevance. [33]. However, the study on the deterioration mechanism of cotton seeds was still rarely reported. In this study, we used an artificial aging treatment on 'Xinluzao 74' cotton seeds to investigate the physiological and biochemical changes during aging, with the aim of providing new insights into the mechanism of cotton seed aging.

## Materials and methods

### Material sources and seed aging treatments

Seeds of cotton (Gossypium hirsutum L.) cultivar "Xinluzao 74", which was kindly provided by Tianzuo Agricultural Science and Technology Development Co., Ltd., Xinjiang, China, were used in the study. The seeds collected were stored at 4°C in tightly sealed containers before experiment. The initial seed moisture content (SMC) of control seeds and seeds aged for 1,2, 3, 4, and 5 days was 9.93%, 9.49%, 9.32% 9.45%, 9.23%, and 10.08%, respectively. The SMC was determined in accordance with international seed testing association (ISTA) [34].

Artificial aging was performed using the method described by Tesnier et al. [35]. Briefly, cotton seeds were placed on the metal mesh, and then hermetically stored in 45°C±1°C, RH 100% for 1, 2, 3, 4, and 5 days, respectively. This condition is considered to reliably simulate the accelerated ageing across oilseed species. The aged seeds were dried back to original water content and kept at 4°C, and the cotton seeds kept in seed storage container in lab (26°C±1°C, RH 30%) were used as controls (CK).

### Germination assays

Germination tests were performed based on the method of ISTA (2015) [36] with slight modifications. The seeds were placed on filter papers moistened with deionized water at 28±1°C in the dark, with 50 seeds in four replications. Seeds were considered germinated when the radical passes through the seed coat and was at least 2 mm in length, and the germination rate was counted every two hours, with the final statistical time of germination rate was 54 h.

## Seed viability evaluation and dehydrogenase activity determination

Seed viability was determined using the TTC (triphenyl tetrazolium chloride) staining experiment according to the method of Li (2003) [37] and slightly modified. The seeds were cut in halves along the hilum, soaked in TTC Staining Solution (0.5%) for three hours at 35°C in the dark, and then photographs were taken under a binocular microscope. The dehydrogenase activity was then determined using an improved TTC staining method [38], the activity of enzyme was expressed in units of "U·g$^{-1}$", with four replicates each sample.

## Determination of seed conductivity

The conductivity of seed soaking solution was determined with reference to the method of Li (2003) [37]. Seeds were soaked in 520 mL deionized water at 20°C for 4, 8, 12, 16, 20, and 24 h, then the conductivity of each sample was measured with a conductivity-meter (mettler toledo InLab730), with four replicates each sample. Results were expressed in units of "µS·cm$^{-1}$·g$^{-1}$".

## Determination of seed MDA content

MDA (malondialdehyde) content in dry seeds and seeds imbibed for 16 h was determined by thiobarbiturate colorimetry [39] in cotton embryos aged for 1, 3 and 5 d, respectively. Briefly, seeds (about 20 g) were homogenized with 10% trichloroacetic acid and centrifuged at 8000 rpm for 20 min, with 4 biological repetitions per treatment, then 2 ml of supernatant was mixed with 2 mL of 0.6% TBA. Then the mixture was incubated at 100°C for 30 min and quickly cooled, the absorbance was measured using a spectrophotometer at 532 nm, 600 nm, and 450 nm.

MDA concentration (µmol·L$^{-1}$) =6.45×(OD532-OD600)-0.56 × OD450. MDA content (nM·g$^{-1}$) = c·x·v·w$^{-1}$ (c: MDA concentration, v: total volume of extract, w: fresh weight of dry seeds or imbibed seeds.) The results were expressed as nM·g$^{-1}$.

## Ultrastructure analysis of seed embryos

The root tip cell ultrastructure was observed using transmission electron microscope (Tecnai G2 Spirit Bio, Thermo Fisher Scientific Inc.), and sections of half-radicle tips were prepared using a standard glutaraldehyde-osmium method.

## Determination of SOD and POD activities

The SOD activity in dry seed embryo was determined using Nitrogen blue tetrazolium photochemical reduction method [40]. POD activity was determined by guaiacol method [40], the enzyme activity was expressed as "U·g$^{-1}$", with four replicates each sample.

## Determination of ·O$_2^-$ production rate and H$_2$O$_2$ content

The production rate of ·O$_2^-$ was detected by a cytochemical method using tetrazolium nitroblue (NBT) [41]. The unit of ·O$_2^-$ production rate is " nmol·g$^{-1}$·min$^{-1}$"(g: fresh weight of dry seeds).

The content of hydrogen peroxide (H$_2$O$_2$) in embryo was determined as described by Sagisaka (1976) [42]. Briefly, seeds were homogenized with 0.1% (w/v) TCA (trichloroacetic acid, containing 10 mM EDTA) and centrifuged at 10,000 rpm for 20 min at 4°C, the supernatant was then mixed with 0.2 mL of 10 mM PBS (phosphate buffered saline, pH 7.0) buffer, and 1 mL of KI (1 M), the supernatant was analyzed at 390 nm. The H$_2$O$_2$ content (µmol·g$^{-1}$) = (C·Vt) · (g·V1)$^{-1}$. C: H$_2$O$_2$ concentration (µmol) in the sample on the standard curve, Vt: total volume of sample extraction solution (ml), V1: the volume of the extracted liquid of the samples (ml), g: fresh weight of seeds (g).

## Determination of respiration rate, ATP content and ATP synthase activity of cotton seeds

The seed respiration rate was determined by the acid-base titration [43], the solutions used were H$_2$C$_2$O$_4$·2H$_2$O and Ba(OH)$_2$, the unit of respiration rate is mg·g$^{-1}$·h$^{-1}$. Briefly, about 20 g of seeds (dry seeds and imbibed seeds) were hung in

a net bag in a sealed glass jar with 15 ml of Ba(OH)$_2$ solution, and then put them in a 28°C constant temperature incubator for 4 h. The blank control was titrated, and the amount of oxalic acid ($V_0$) was determined, and then the amount of oxalic acid ($V_1$) was determined until the solution changed from pink to colorless. Respiration rate (mg·g$^{-1}$·h$^{-1}$) =($V_0$-$V_1$) ·seed weight (g)$^{-1}$·reaction time (h)$^{-1}$

The ATP content was determined using an ATP Content Assay Kit (WST-1 Method) (AKOP004U, Beijing Boxbio Science & Technology Co., Ltd.) according to the manufacturer's instruction.

The activity of ATP synthase of seed embryo was determined by plant ATP synthase enzyme-linked immunoassay kit (HB356X-Pt, Shanghai Hengyuan Biotechnology Co., Ltd.) according to the manufacturer's instruction, the activity of ATP synthase was expressed in units of " U·g$^{-1}$", and the buffer solution used here is PBS (phosphate buffer solution, 0.01 M, pH 7.4).

### ATP synthase subunit mRNA integrity detection

Total RNA of cotton seed embryo was extracted using TRIzol Plant kit (ET121, Beijing Quanjin Biotechnology Co., Ltd.). The integrity of the RNA obtained was analyzed via 0.1% agarose gel electrophoresis, and the quantification of RNA was determined using the NanoDrop2000 (Thermo Fisher Scientific, USA). The cDNA synthesis was carried out using the Fast Quant RT Kit (with gDNase) (TianGen, Beijing). The gene sequence of ATP synthase subunits (α, β, γ, δ and ε) in *Gossypium hirsutum* was searched through NCBI website, and primers were synthesized by Shanghai Shenggong Co., Ltd. Primer sequences used for qRT-PCR analysis are listed in S1 Table. For each sample, three biological replicates and three technical replicates were performed.

The integrity of mRNA was analyzed by reverse transcription blocking-double primer amplification. qRT-PCR detection was performed on a Bio-Rad iQ5 Real-Time PCR System (Bio-Rad, United States) using TransStrat Top G reen qPCR Supermix Kit (TransGen), as described by Ma et al. (2013) [44]. The amplification steps were: 95°C for 3 min; 95°C for 10 s, 59°C for 30 s, 72°C for 30 s, 40 cycles, and 72°C for 4 min. The relative expression of the genes was calculated using the 2$^{-\triangle\triangle Ct}$ analysis method, and the integrity of the mRNAs was represented as *R*-value.

$$R = 2^{-\triangle Ctx}/2^{-\triangle Ct_0}$$

### Data analysis

The data were calculated and analyzed using Microsoft Excel 2010. Data were subjected to one-way ANOVA followed by Dunnett's test (SPSS 20.0). Pearson correlation evaluation and heat map between traits were performed in Origin 2022($P \leq 0.05$).

## Results

### Seed germination rate

The germination performance of seeds aged for 1 d, 2 d, 3 d, 4 d, and 5 d were determined, respectively, and the result indicated that the seed germination rate gradually decreased as aging progressed (Fig 1, Table A in S4 Table). A small number of the control seeds (CK) began germinating after 14 hours of imbibition, with the germination rate reaching 86.0% at 34 hours and a maximum of 94.7% at 46 hours. Seeds aged for 1 d, or 2 d had a small number of seed germination at 18 h, and the germination rate reached 85.6% or 73.9% after imbibition for 34 h, respectively. Seeds aged for 3 d, 4 d, or 5 d had no seed germination at 18 h, and the germination rate reached 60.4%, 22.5% or 1.8% after imbibition for 34 h, respectively. The final germination rates were 73.3%, 41.3%, 14.0%, respectively, which were significantly lower than those of the control and seeds aged for 1 d. These results showed that artificial aging treatment can lead to a decline in seed germination rate.

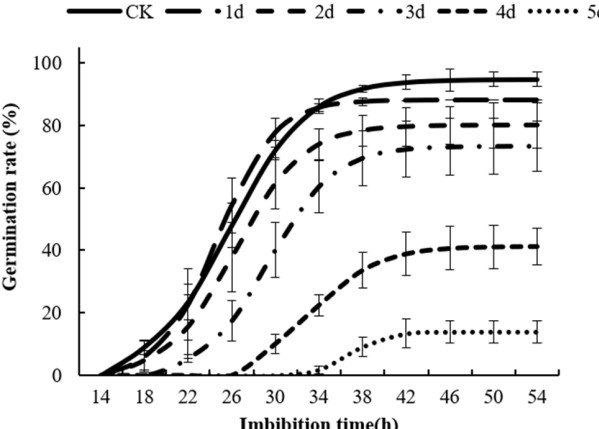

**Fig 1. Germination performance of different aged seeds.** All values in the figure are represented as "mean ±SD" (error bars), the same below.

## Performance of different aged seed viability

The seed viability was determined using TTC staining method. The results showed that seed embryos of the control and those aged for 1 day were stained red (Fig 2A, 2B). Seed embryos aged for 2 days stained weakly red, particularly in the embryonic axis (Fig 2C). The radicle staining in seeds aged for 3 and 4 days was significantly weaker than that of the control seeds, with only the root tip staining a light red (Fig 2D, 2E). Embryos of seeds aged for 5 days showed no staining (Fig 2F). These results showed that artificial aging treatments lead to a decline in viability of seed embryos, and there were some differences in the responses of different organs of seed embryos to aging treatment.

Dehydrogenase is an essential enzyme in the process of seed respiration, and its activity is closely related to the seed viability. The results showed that the dehydrogenase activity gradually declined as the duration of artificial aging treatment increased (Fig 3, Table B in S4 Table). The dehydrogenase activity of control seeds was 4.66 U·g$^{-1}$, cotton seeds after aging treatment showed significantly lower activity, among which the dehydrogenase activity of cotton seeds aged for 5 d is 2.90 U·g$^{-1}$, which was significantly reduced by 37.80% compared with the control.

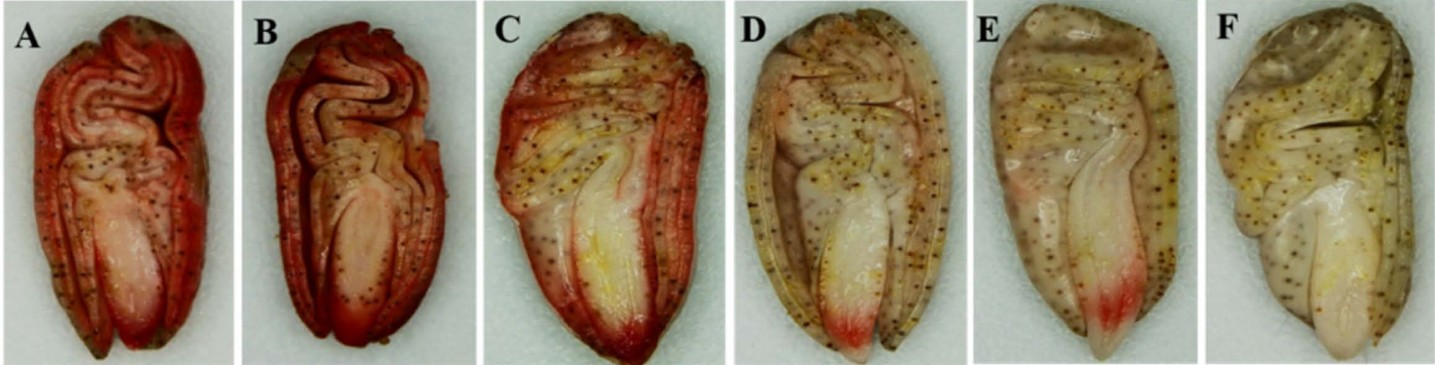

**Fig 2. TTC staining of different aged cotton seed embryos. (A)** CK. **(B)** seeds artificially aged for 1d. **(C)** seeds artificially aged for 2d. **(D)** seeds artificially aged for 3d. **(E)** seeds artificially aged for 4d. **(F)** seeds artificially aged for 5d.

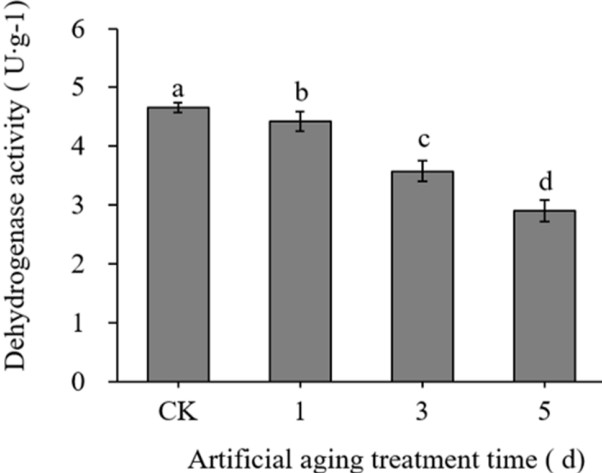

**Fig 3. Effects of different artificial aging treatments time on the dehydrogenase activity of cotton seeds.** Significant differences between treatment groups and control group are indicated as determined by Dunnett-t test, and the lowercase letter above the histograms represents a significant level of *P≤ 0.05* according to Duncan test., the same below. Means with similar letters in each column are not significantly different at 5% level.

## Plasma membrane integrity assessment

The conductivity values of the seeds aged for 1, 2, 3, 4, and 5 d were determined, respectively. The results showed that the conductivity of all seeds showed an increase with soaking time prolonged (Fig 4, Table C in S4 Table). After soaking for 4 h, 12 h, and 24 h, the conductivity of control seeds was 10.90 $\mu S \cdot cm^{-1} \cdot g^{-1}$, 15.27 $\mu S \cdot cm^{-1} \cdot g^{-1}$ and 18.49 $\mu S \cdot cm^{-1} \cdot g^{-1}$, respectively. In seeds aged for 1 day, conductivity was 13.57 $\mu S \cdot cm^{-1} \cdot g^{-1}$, 18.17 $\mu S \cdot cm^{-1} \cdot g^{-1}$, and 22.13 $\mu S \cdot cm^{-1} \cdot g^{-1}$, respectively; In seeds aged for 5 days, conductivity was 20.46 $\mu S \cdot cm^{-1} \cdot g^{-1}$, 30.73 $\mu S \cdot cm^{-1} \cdot g^{-1}$ and 42.17 $\mu S \cdot cm^{-1} \cdot g^{-1}$, respectively. The conductivity of aged seeds was significantly higher than that of the control. These results indicated that the artificial aging treatment caused the loss of membrane integrity of seed embryos and serious leakage of contents.

## MDA content

Malondialdehyde (MDA) have been identified as an effective marker of lipid peroxidation. The MDA content in cotton seeds aged for 1, 3, and 5 days was determined. Results showed that control seeds had the lowest MDA content (0.55

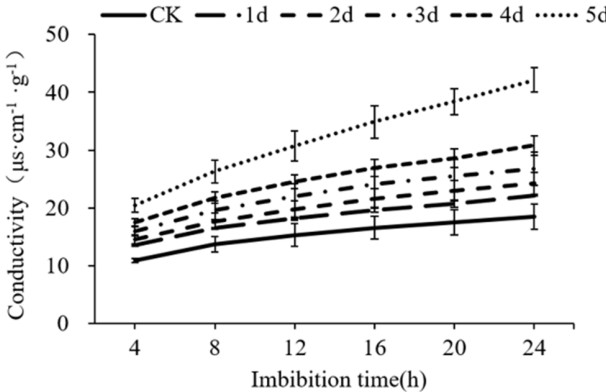

**Fig 4. Effect of different aging treatments on the conductivity of cotton seeds.**

nM·g$^{-1}$), with a noticeable accumulation of MDA in aged seeds (Fig 5, Table D in S4 Table). The content of MDA in dry seeds aged for 1 d, 3 d, and 5 d were 0.68 nM·g$^{-1}$, 0.82 nM·g$^{-1}$ and 1.14 nM·g$^{-1}$, respectively, which increased by 23.36%, 49.32%, and 107.10%, respectively, compared with the control. After imbibition for 16 hours, the MDA contents of seeds aged for 1 d, 3 d and 5 d were 0.18 nM·g$^{-1}$, 0.51 nM·g$^{-1}$ and 1.06 nM·g$^{-1}$, respectively, which were significantly increased by 27.60%, 262.42% and 648.34% compared with the control (0.14 nM·g$^{-1}$). The MDA content of all seeds decreased after imbibition, but the degree of declines was significantly different, the more severe the aging treatment, the smaller the decrease in MDA content in seeds, indicating that the oxidation of lipids was inhibited during imbibition.

### Cell ultrastructure of cotton seed embryos

We observed the root tip cell ultrastructure of the control and 3-day aging seed by transmission electron microscope (Fig 6): the nuclear contour of CK (Fig 6A) is regular, with a homogeneous matrix, spherical nucleoli, and no chromatin clumping. Internal mitochondria have uniform structure, clear membrane profile, differentiation with a few of cristae, and sparse matrix (Fig 6B); lipid droplets are distributed outside the cytoplasm, and the cell membrane and cytoplasm structure are clearly visible. After 3 days of aging (Fig 6C), the nuclear profile had become irregular, and clumped chromatin was evident, with irregularly dispersed, dense clumps of chromatin. Lipid droplets were disposed both along the cell periphery and within the cytoplasm. Zoomed-in view of the seed that aged for 3 days (Fig 6D) shows that mitochondria presented generally circular profiles and are markedly swollen, showing derangement of the internal membrane system, the distortions of the outer membrane and/or cristae ranging from becoming disorganized to being almost absent, deteriorated mitochondria, and showing derangement of the internal membrane system. The results of ultrastructure observation of cotton seeds show that artificial aging treatment damaged the mitochondrial structure and cell nuclear structure of root tip cells, which may lead to cytogenetic damage and damage to the mitochondrial energy metabolism system.

### Determination of ·O$_2$- production rate and content of H$_2$O$_2$ of cotton seed embryos

The effect of artificial aging treatment on the ·O$_2$- production rate (Fig 7A, Table E in S4 Table) and H$_2$O$_2$ content (Fig 7B, Table E in S4 Table) of seed embryos were determined. The results showed that the ·O$_2$- production rate in control seed embryos was 8.35 nmol·g$^{-1}$·min$^{-1}$, and the ·O$_2$- production rate increased significantly during aging treatment, reaching a peak after aging for 3 d (16.44 nmol·g$^{-1}$·min$^{-1}$), then declined after aging for 5 d (12.13 nmol·g$^{-1}$·min$^{-1}$), but it was still significantly higher than that of the control (Fig 7). The ·O$_2$- production rate of seeds aged for 1 d, 3 d, 5 d increased by 59.80%, 96.86% and 45% compared with that in control, respectively.

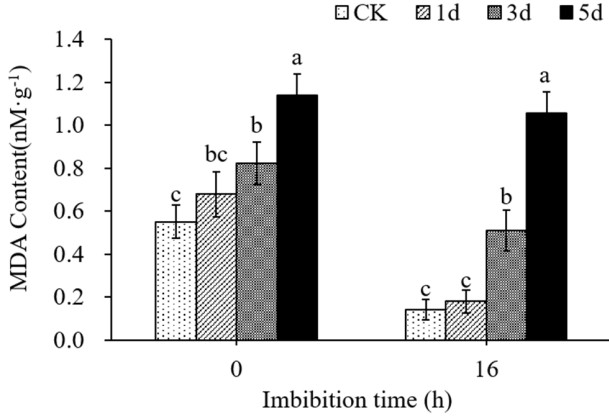

**Fig 5. MDA content in cotton seed.**

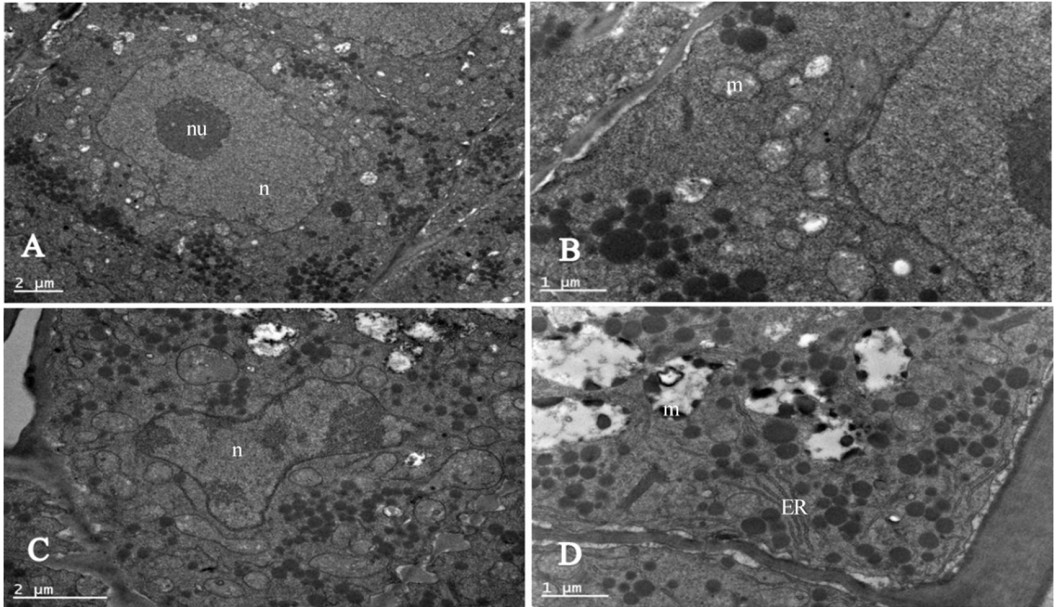

**Fig 6. Mitochondrial structure of root tip cells of cotton seeds treated by artificial aging.** Scale bars specifying actual dimensions in images. **(A)** CK seed. **(B)** CK seed displayed in a zoomed-in view. **(C)** Seed aged for 3 days. **(D)** Zoomed-in view of the seed that aged for 3 days. n: nucleus; nu: nucleolus; m: mitochondria; ER: endoplasmic reticulum.

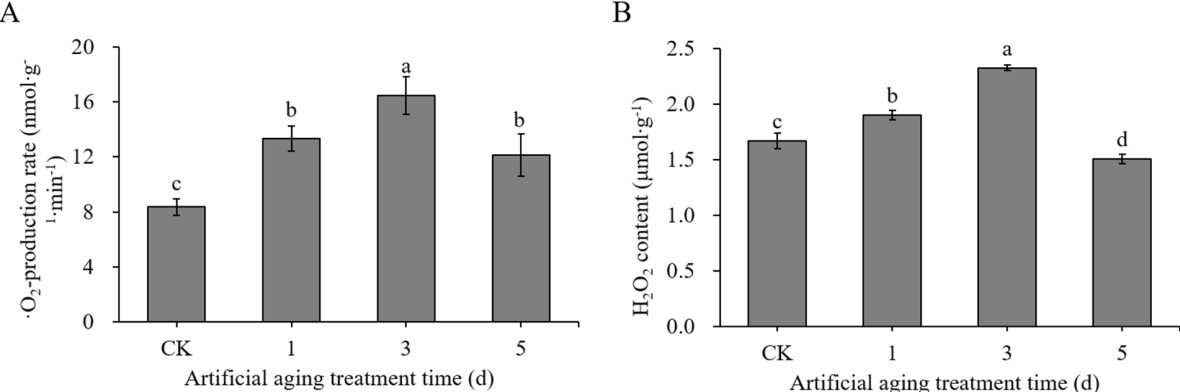

**Fig 7. $\cdot O_2^-$ production rate and $H_2O_2$ content in different artificial aging treatments. (A)** $O_2^-$ production rate. **(B)** $H_2O_2$ content.

The content of $H_2O_2$ showing a trend of increase first and then decrease (Fig 7B), it was 1.67 µmol·g$^{-1}$ in the control, and after aging for 1 d or 3 d, it increased significantly to 1.90 µmol·g$^{-1}$ or 2.33 µmol·g$^{-1}$, respectively, representing a 13.82% and 39.32% increase compared with that in control. After aging treatment for 5 d, the content of $H_2O_2$ was 1.51 µmol·g$^{-1}$, which dropped significantly compared with that in control.

### Determination of SOD and POD activity

Antioxidant enzymes (SOD and POD) are important factors in the enzymatic system for scavenging ROS. The determination results of SOD activity showed that in dry seeds, the enzyme activity in control was the highest (237.04 U·g$^{-1}$), and

the SOD activity decreased gradually as the degree of aging increases (Fig 8, Table F in S4 Table). After aging treatment for 1 d, 3 d, or 5 d, the SOD activities were 206.63 U·g⁻¹, 160.26 U·g⁻¹, or 89.25 U·g⁻¹, respectively, showing reductions of 12.83%, 32.39%, and 62.35% compared with that in the control. After imbibition for 16 h, the SOD activity of seeds increased significantly, but the trend of enzyme activity is still decreasing significantly with the extension of aging time. The SOD activity of cotton seeds aged for 1 d, 3 d and 5 d were 257.89 U·g⁻¹, 228.22 U·g⁻¹, and 146.35 U·g⁻¹ respectively, which decreased by 4.93%, 15.86% and 46.05% respectively, compared with the control. The trend of POD activity (Fig 8B) is similar to SOD activity (Fig 8A). These results showed that artificial aging treatment damaged the antioxidant enzyme system of embryos, and reduced the activities of SOD and POD.

### Effects of different artificial aging treatments on energy metabolism (respiration rate, ATP content and ATP synthase activity) of embryo

Respiration activity is intimately connected to the metabolism of seeds, and the ATP produced by respiration is the essential energy source for seed germination. The respiration rate is a crucial indicator of metabolic activity. The effect of aging treatment on the respiration rate in seed germination was determined. The results showed that the respiration rate of dry seeds was very weak in both control (0.0108 mg·g⁻¹⁻ʰ) and different aging treatments (0.0083 mg·g⁻¹⁻ʰ, 0.0058 mg·g⁻¹⁻ʰ, 0.0033 mg·g⁻¹⁻ʰ represented aging treatment 1 d, 3 d and 5 d, respectively) (Fig 9, Table G in S4 Table). The respiration rate of all seeds increased significantly with imbibition, however, the magnitude of increase varied. The control seeds experienced the most significant increase in respiration rate, while seeds aged for 3 d and 5 d only showed a slight rise. After 16 hours of imbibition, the respiration rate in control seeds was 0.34 mg·g⁻¹·h⁻¹, while that in the seeds aged for 1 d, 3 d and 5 d were 0.30 mg·g⁻¹⁻ʰ, 0.25 mg·g⁻¹⁻ʰ and 0.20 mg·g⁻¹⁻ʰ, respectively, which were significantly reduced by 9.46%, 26.74% and 39.61% compared with the control, respectively. This indicates that the artificial aging treatment led to a decrease in the respiration rate of the seeds during germination.

ATP produced by respiration metabolism is an important energy source for life activities, and the ATP content will be affected inevitably when the respiration intensity of seed embryo decreases. The results showed that the ATP content in dry seeds was very low (Fig 10A, Table H in S4 Table), and there was no significant difference between aged-treated seeds and control seeds. With the extension of imbibition time, ATP content gradually increased in all seed embryos, but the extent of the increase varies. After imbibition for 16 hours, the contents of ATP in seeds aged for 1 d, 3 d, 5 d were 0.93 μmol·g⁻¹, 0.69 μmol·g⁻¹ and 0.63 μmol·g⁻¹, respectively, which is significantly reduced by 7.49%, 30.57%, and 37.33% compared to the control (0.98 μmol·g⁻¹).

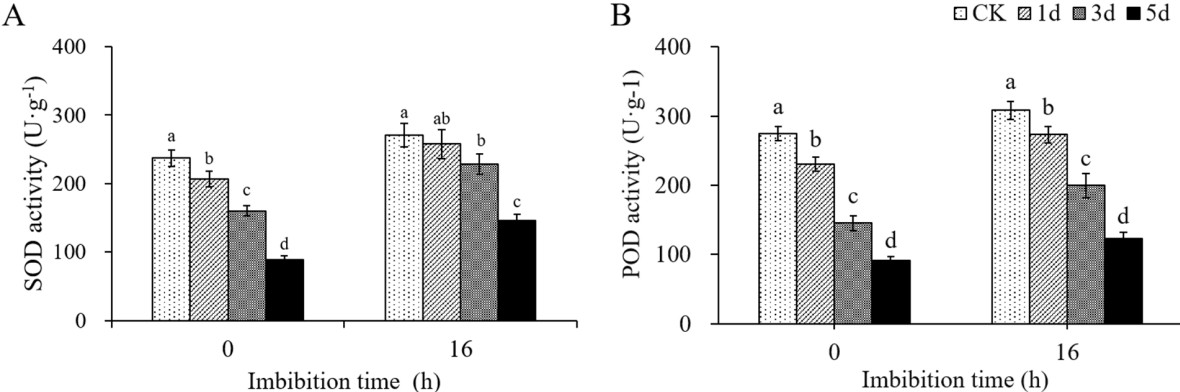

**Fig 8. Effects of artificial aging treatment and imbibition on SOD and POD activity. (A)** SOD activity. **(B)** POD activity.

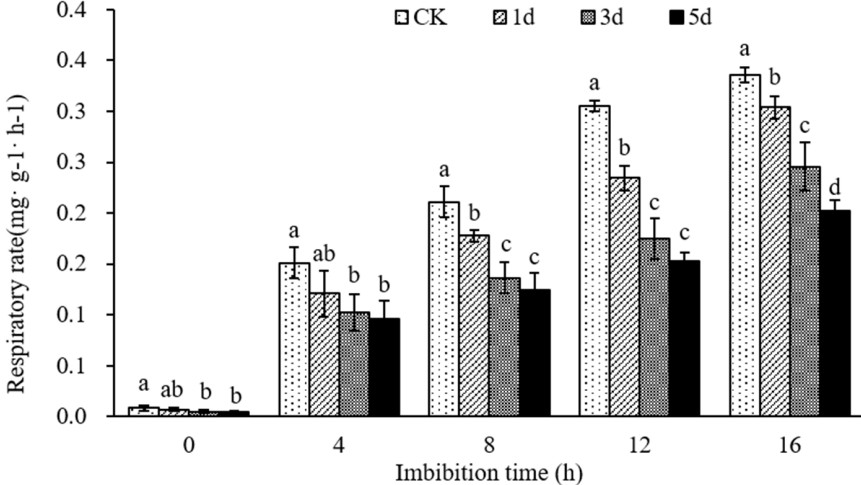

**Fig 9. Changes of respiration rate during seed germination under different artificial aging treatments.**

The ATP content in cotton seed embryos decreased above mentioned would possibly be due to the influence of ATP synthase activity. The activity of ATP synthase in seeds was measured (Fig 10B, Table H in S4 Table), the results showed that the activity of ATP synthase in dry seed embryos of both control and aged seeds was very low. The enzyme activity in control was 210.3 U·g⁻¹, while in seeds aged for 1, 3, and 5 d, the enzyme activity was 199.23 U·g⁻¹, 190.92 U·g⁻¹, and 171.8 U·g⁻¹, respectively, showing a significant decrease. This indicates that aging treatment has affected the ATP synthase activity in dry seeds. After imbibition, the activity of ATP synthase in seed embryos generally increased, but with significant differences in increasing extent. After imbibition for 16 hours, the enzyme activity in control was 837.6 U·g⁻¹, and that in the embryos aged for 1 d, 3 d and 5 d were 782.72 U·g⁻¹, 714.29 U·g⁻¹ and 669.4 U·g⁻¹, respectively, significantly reduced by 6.55%, 14.72% and 20.08% compared with the control. These results indicate that artificial aging treatment affects seed respiration metabolism and consequently reduces ATP synthesis.

## ATP synthase subunit mRNA integrity analysis

The ATP synthase is composed of five subunits: α, β, γ, ε, and δ. Modification or damage to the mRNA encoding these subunits can affect their structure and function, thereby impacting the activity of ATP synthase in seeds. The integrity of

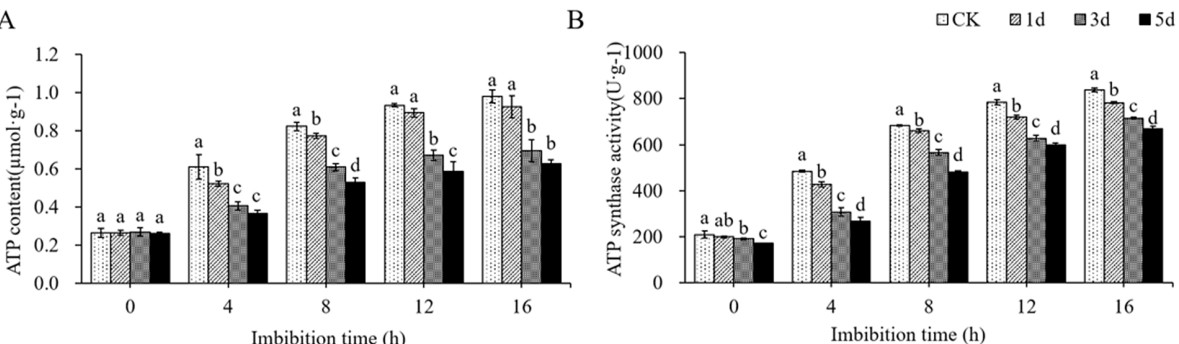

**Fig 10. Changes of ATP content and ATP synthase activity (B) during imbibition of seeds treated with artificial aging. (A)** ATP content. **(B)** ATP synthase activity.

these 5 genes in seeds aged for 3 d was analyzed by reverse transcription blocking-double primer amplification method, The relative expression levels of the 5' and 3' ends of each subunit are listed in the S2 Table, the integrity of the mRNAs are represented as *R-value* (listed in the S3 Table). The results (Fig 11) showed that the *R* value of embryo *β* subunit mRNA increased significantly (1.18) after aging for 3 d, which was 18% higher than the control (1.00), indicating a potential abnormal degradation at the 5'-end. The *R* values of other subunits decreased to different degrees, among which the *R* value of *ε* subunit mRNA was 0.88, showing a slight decrease. Meanwhile, the *α, γ,* and *δ* subunits' R values of mRNA were 0.32, 0.24, and 0.80, respectively, showing significant decreases compared with the control by 67.88%, 75.66%, and 20.19%, which suggests varying degrees of degradation at the 3'-end of these three subunit mRNAs. These results demonstrates that the artificial aging treatment caused damage to the mRNA of ATP synthase subunits, which in turn affected the subunits' synthesis quality and efficiency, and ultimately influencing the activity of ATP synthase in seeds.

## Correlation analysis

Pearson correlation coefficient (when $P \leq 0.05$) between ROS (the generation rate of $\cdot O_2^-$ and the $H_2O_2$ content) and physiological and biochemical alterations of seeds after artificial aging treatment for 1 day and 3 d, as well as the integrity of ATP synthase subunit mRNA was analyzed. The results demonstrated a negative correlation between the $H_2O_2$ content and the POD and SOD activity of seeds (Fig 12, Table I in S4 Table), indicating that the decrease in antioxidant enzyme activity in artificial aged seed embryos would be an important factor for ROS accumulation. The $H_2O_2$ content is significantly negatively correlated with the germination rate and respiration rate of cotton seeds, suggesting that $H_2O_2$ might be the primary factor for reducing the germination ability and respiratory metabolism of cotton seeds. Furthermore, the *R* values of the *α, γ, δ,* and *ε* mRNA of ATP synthase subunits. These results suggest that the integrity of these five subunits is closely related to the content of reactive oxygen species, and that an excessive generation of ROS would be the primary factor responsible for the damage to the mRNAs of ATP synthase subunit.

## Discussion

Substantial evidence indicates that ROS plays a significant role in seeds: at low concentrations, they regulate physiological processes such as seed germination and dormancy [45,46], while excessive accumulation induces oxidative damage and reduces seed vigor [47]. The overproduction of ROS ($\cdot O_2^-$, $H_2O_2$, and $\cdot OH$) generation, a decline in antioxidant capacity, and a slow build-up of oxidative damage are considered major drivers of seed deterioration. And in most cases, the levels of $\cdot O_2^-$ and $H_2O_2$ are demonstrated to be associated with the dysfunction of mitochondria complex I [48]. The antioxidant system serves as a critical defense mechanism against ROS during seed aging [20,49]; In deteriorated seeds,

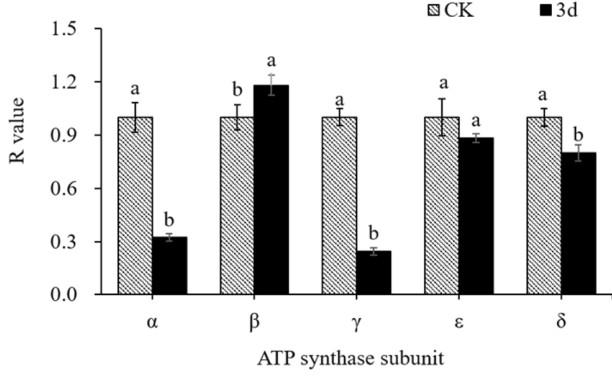

**Fig 11. Integrity of ATP synthase subunit mRNA during aging.**

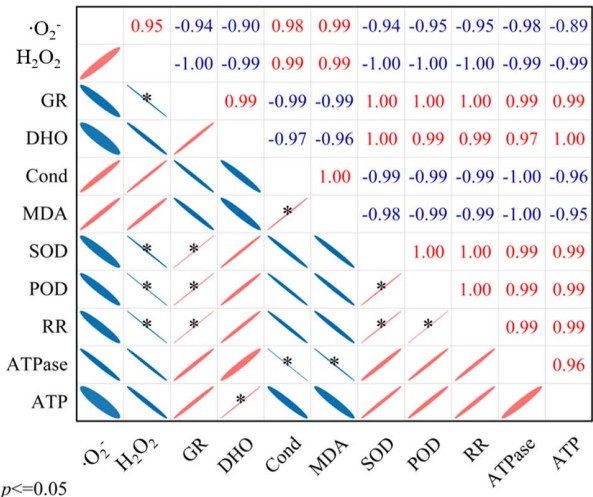

* $p<=0.05$

**Fig 12. Correlation analysis between physiological and biochemical alterations of seeds with ROS.** Pearson correlation between physiological traits and cluster analysis of physiological traits in seeds through heatmap plot. *. Correlation is significant at the 0.05 level. GR: germination rate; DHO: dehydrogenase activity; Cond: electrical conductivity; RR: respiration rate; ATPase: ATP synthase activity; ATP: ATP content.

antioxidant enzymes were inactivated, leading to excessive accumulation of ROS, which may cause oxidative damage to several biological macromolecules and disrupt cell metabolism, thus causing physiological imbalance [50,51]; It has been found that during the aging process of wheat seeds, the excessive accumulation of ROS-induced protein carbonylation in the embryo is the main reason for the reduction in seed viability [19]; The decrease in seed germination rate of stored beech seeds was significantly correlated with the increase in ROS ($H_2O_2$) levels, and proposed that ROS is a major factor in the loss of seed viability [47].

Seed deterioration can be classified into three phases [6]: the first phase was the initial stage, in which Amadori and Maillard reactions occur within the seed, causing a slight decline in antioxidant enzyme activity and minor genetic damage [52–54], with little effect on seed vigor. In the second stage, oxidative damage to cells became apparent due to the decrease of antioxidant enzyme activity, lipid peroxidation and malondialdehyde production were driven by excessive reactive oxygen species (ROS), which resulted in additional damage to genetic material and a considerable decline in seed vigor [50]. In the third stage, mitochondrial membrane lipid peroxidation leads to metabolic dysfunction, which in turn promotes additional toxic compounds like ROS and malondialdehyde (MDA). This vicious circle exacerbates genetic damage and ultimately results in complete loss of seed viability.

SOD, a key ROS-scavenging enzyme in plants and animals, can catalyze the dismutation of $O_2^-$ into $H_2O_2$ [48]. POD is an antioxidant enzyme that directly oxidizes phenolic or amine compounds with $H_2O_2$ as an electron acceptor, and thereby detoxifying both $H_2O_2$ and phenolic amines [55]. In this study, prolonged artificial aging led to a sharp decline in the activities of SOD and POD in cotton seeds (Fig 8), while $H_2O_2$ content peaked after 3 days of aging (Fig 6). These trends are consistent with observations in covered oat and naked oat (*Avena nuda* L.) seeds [56], wheat [57], soybean [58] and *Allium mongolicum* Regel seeds [59]. The MDA content in dry seeds increased significantly, and decreased significantly in control seeds after imbibition for 16 h, but it was not significant in aged seeds—suggesting impaired antioxidant repair capacity and aggravated lipid peroxidation in aged seeds. Microscopic examination further revealed substantial damage to the cell membrane, mitochondria, and chromatin structure in seed embryos (Fig 6), aligning with earlier reports [13], indicating that ROS is a key factor causing damage to a series of active components and structures in cells. These results are consistent with theoretical model proposed by Ebone [6].

Mitochondria are the main site of ROS generation and play a crucial role in plant cell redox homeostasis and signaling. However, the mitochondrial antioxidant system is relatively weak, rendering mitochondria particularly vulnerable to oxidative attack [13], this further accelerates the process of seed deterioration [60,61]. During seed aging, the antioxidant defenses (antioxidant enzymes and other antioxidants) in mitochondria can be overwhelmed, leading to oxidative damage that accelerates seed deterioration [62]. For example, NADH dehydrogenase (Complex I; NDH) is a critical site of $\cdot O_2^-$ production, but this superoxide generation could induce self-inactivation with specific protein radical formation, it has been demonstrated that several domains of mitochondria are susceptible to oxidative attack, and their oxidative modification results in decreased electron transfer activity [63]. The production and accumulation of ROS in mitochondria damages mitochondrial ultrastructure, accompanied by the down-regulation of key proteins such as ATP synthase and malate dehydrogenase, and disrupts metabolic pathways like the TCA cycle and electron transfer chain on the mitochondrial membrane. These defects collectively impair ATP production and energy metabolism [10].

ATP synthase, located on the mitochondrial inner membrane, catalyzes over 90% of cellular ATP and is essential for energy metabolism [64]. ATP synthase is composed of several subunits, among which $F_1$, which is composed of five subunits, α, β, γ, δ and ε, is situated between the matrix and the inner membrane of the mitochondria. The β subunit is the catalytic site of ATP synthase, while the γ subunit is the central axis of the catalytic site, which driven by proton energy to rotate and then catalyze ATP synthesis [65,66]; These subunits play an important role in energy metabolism by converting ADP into ATP in the presence of a transmembrane proton gradient. Recent studies on plant indicate that damage to the mRNA encoding these subunits might cause ATP synthase dysfunction, affecting mitochondrial metabolism [67], for example, Lapaille et al [68] have proved that the β subunit plays an important role in ATP synthesis. A decline in ATP synthase activity in seeds reduces ATP content, ultimately impairing seed germination by diminishing energy production and material supply [25]. Mitochondria are a key factor in seed aging [60]. For instance, in aged pea seeds, the respiratory rate decreases, accompanied by a decline in ATP production and reduced efficiency of electron transport [69].

Our findings revealed a significant increase in malondialdehyde (MDA) content (Fig 5), an indicator of lipid peroxidation. Correspondingly, we observed that the mitochondria of the radicle apical cells were severely degenerated, showing the internal matrix and membrane system disorder, and indicating potential mitochondrial dysfunction (Fig 6). This phenomenon is correlated with a decline in key physiological parameters, including respiration rate, ATP synthase activity, and ATP content. We also discovered that the $H_2O_2$ content was negatively correlated with the germination rate and respiration rate of cotton seeds (Fig 12). These results indicated that the mitochondrial metabolic dysfunction caused by ROS is an important reason for the decrease of cotton seed vigor, which in good agreement with the study of seed deterioration of other oil crops like maize [13], safflower [21] and soybean [58].

During seed development, large quantities of mRNA are accumulated, which are not only essential for maintaining seed viability and lifespan, but also regulates physiological activities such as dormancy and seed germination [70]. ROS-mediated mRNA oxidation may cause nucleic acid strand breaks or adducts formation, which disrupts translation, causes errors in protein translation, or reduce the efficiency of protein synthesis [71,72], ultimately affecting seed vigor. It has been shown that the expression of mRNA encoding antioxidant enzymes is related to seed longevity. During the senescence of pea (Pisum sativum L.) seeds, the mRNA related to oxidative stress (POD, SOD, etc.) was down-regulated, and the antioxidant capacity of seeds decreased [73]. The targeted oxidation of mRNA during the after-ripening period of sunflower seeds regulates the relaxation and release of dormancy, indicating that mRNA oxidation plays an important role in regulating seed dormancy and germination [74,75]. The RNA integrity number (RIN), a metric of RNA quality, is strongly correlated with germination capacity, suggesting that the non-enzymatic RNA fragmentation might be a key mechanism in seed aging [76,77].

In our study, the deformation of the nucleus and chromatin in seed embryos was clearly observed, indicating genetic damage in aged seeds (Fig 6). The chromatin of seed embryos was seriously damaged. We investigated the relationship between ATP generation and seed vigor in cotton seeds, with a particular focus on the role of ROS in ATP synthase

dysfunction. More importantly, we identified oxidative damage to mRNAs encoding ATP synthase subunits as a central mechanism in cotton seed deterioration. The integrity of α, β, γ, δ, and ε subunit mRNAs declined with aging (Fig 11), and this loss correlated with ROS accumulation—suggesting ROS-induced oxidation [77]; Among these five subunits, the α and γ subunits were the most severely damaged. Damage to the α subunit's mRNA might destroy the catalytic site of ATP synthase. Damage to the γ subunit may impair ATP synthase's ability to utilize proton energy for ATP synthesis, ultimately reducing ATP production. These results indicate that ROS overaccumulation causes oxidative damage to the mitochondrial ATP synthase subunit mRNA, leading to subunit defects and ATP synthase dysfunction. The data presented in Fig 5 illustrate that increased MDA levels correspond to decreased respiration rate and ATP synthase activity. This decline is critical because ATP is essential for seed metabolic processes, and its reduction indicates impaired energy metabolism. Furthermore, our analysis revealed a negative correlation between $H_2O_2$ levels and both germination and respiration rates (Fig 12). Increased $H_2O_2$ content may lead to oxidative stress, resulting in loss of seed viability. Consequently, persistent decreases in ATP production and energy supply during seed germination may be the main reason for reduced seed vigor. We therefore propose that persistent energy shortfalls during germination, caused by ROS-induced mRNA damage and mitochondrial dysfunction, are a major contributor to seed aging. Our study highlights mRNA oxidation—particularly in genes critical to energy metabolism—as a key mechanism of seed deterioration. The integrity of such mRNAs may serve as a valuable biomarker for predicting seed aging and storage potential.

## Conclusion

The seed aging process is inevitable, affecting all seeds, even under optimal storage conditions. Global warming poses additional challenges, threatening seed viability, crop yields, and germplasm resources. This study reveals that artificial aging in cotton seeds leads to a decline of antioxidant activity and an increase in ROS levels. These changes damage ATP synthase subunit mRNAs, hindering ATP production and reducing seed germination. In conclusion, we believe ROS-induced ATP synthase mRNA degradation is a key molecular event contributing to seed aging. This study provides a novel perspective on seed aging by linking oxidative stress with mitochondrial energy metabolism dysfunction. These findings enhance our understanding of seed deterioration mechanisms and provide insights for seed quality assessment and germplasm preservation strategies.

## Supporting information

**S1 Table. Double primers used for gene integrity detection.**
(PDF)

**S2 Table. Relative expression levels of the 5' and 3' ends of each subunit.**
(PDF)

**S3 Table. R value.**
(PDF)

**S4 Table. Dataset.**
(XLSX)

## Author contributions

**Formal analysis:** Ci Song, Zhenzhen Xing.

**Investigation:** Junming Zhang.

**Methodology:** Ci Song, Fengxu Gu.

**Supervision:** Junying Chen.

Writing – original draft: Ci Song.

Writing – review & editing: Junying Chen.

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
