## [Decision Letter · Decision Letter 0]

1 Oct 2025

Dear Dr. Chen,

Thank you for submitting your manuscript to PLOS ONE. After careful consideration, we feel that it has merit but does not fully meet PLOS ONE’s publication criteria as it currently stands. Therefore, we invite you to submit a revised version of the manuscript that addresses the points raised during the review process.

We look forward to receiving your revised manuscript.

Kind regards,

Haitao Shi

Academic Editor

PLOS ONE

Journal Requirements:

Reviewers' comments:

Reviewer's Responses to Questions

**Comments to the Author**

1. Is the manuscript technically sound, and do the data support the conclusions?

Reviewer #1: Partly

Reviewer #2: Yes

2. Has the statistical analysis been performed appropriately and rigorously?

Reviewer #1: Yes

Reviewer #2: Yes

3. Have the authors made all data underlying the findings in their manuscript fully available?

Reviewer #1: Yes

Reviewer #2: Yes

4. Is the manuscript presented in an intelligible fashion and written in standard English?

Reviewer #1: No

Reviewer #2: Yes

Reviewer #1: I have given my deep reading to the manuscript tilted as ROS production and loss of ATP synthase subunit mRNA integrity as a mechanism of artificial deteriorated cotton seedshave the following concerns regarding the manuscript

1) Can you please reconsider the title with a unique word as it seems so simple and less attractive than the audience

2) Abstract and conclusion are pessimistic as very simple language is used no novelty or significance of work is properly mentioned. I am always very curious about conclusion as it provided crux of whole documents please refine your English in this section.

3) In continuation to my previous comment Please double check grammar as overall language of the paper needs to verify. You can get help through software.

4) Please rewrite your introduction or add constructive information to make it attractive for readers

5) Can you please add graphical/ general picture of your work along abstract to make it clear graphical or tabular representation to make finding easy t access for readers and it usually enhances beauty of the paper. I do like to check these first whenever reading a paper. That’s what I believe, maybe other reviewers think differently.

6) Graphs are very small and very hard to understand. Please make them suitable size, font and titles

7) Surprisingly results are acceptable form but please add more justification in your discussion part and add new findings relevant to you work

8) In Reference sections please add all required details. Make sure you are following the journal pattern overall.

9) Good luck

Reviewer #2: The manuscript addresses an important and underexplored aspect of cotton seed deterioration by linking ROS accumulation with ATP synthase subunit mRNA damage. The study is strengthened by its integration of physiological, biochemical, and ultrastructural analyses, providing a multifaceted view of seed aging. The correlation between ROS levels and mRNA integrity is particularly novel and has potential implications for seed quality preservation. However, the manuscript cannot be accepted in its present form. The authors should address the following points.

1. The rationale for choosing the cultivar "Xinluzao 74" has been briefly stated, but its relevance to broader cotton varieties worldwide could be explained more clearly.

2. Please provide the justification for selecting 45°C and 100% RH could be better supported with references or comparative discussion.

3. In case of Figures 1–3, the figure legends could be expanded to ensure they stand alone without the need to refer to the text.

4. The intensity of TTC staining could be quantified by using image J or similar software if possible.

5. Conductivity data should be discussed for the involvement of any possible non-aging factors influencing conductivity (e.g., seed coat permeability differences).

6. The ROS measurements show the initial decline in H₂O₂ after 5 days of aging should be more carefully interpreted, as alternative explanations beyond ROS scavenging could exist.

7. Regarding the results of respiration and ATP synthase,more explanation is needed regarding whether the observed decreases are cause or consequence of mRNA damage.

8. The discussion section occasionally reiterates results rather than offering deeper mechanistic insights, particularly in the first two paragraphs. The discussion section should be improved please.

9. Minor language issues (e.g., “seriously damaged” on p.6) could be revised for precision and to maintain a formal scientific tone throughout.

**Do you want your identity to be public for this peer review?** For information about this choice, including consent withdrawal, please see our Privacy Policy

Reviewer #1: No

Reviewer #2: **Yes:** Muhammad Amjad Ali

---

## [Author Response · Author response to Decision Letter 1]

22 Nov 2025

We sincerely thank the editor and reviewers for their careful reading and constructive comments. We have carefully revised our manuscript according to all suggestions. Below, we provide detailed responses point by point, with modifications highlighted in the revised manuscript and summarized with artificial line numbers.

Response to Reviewer 1

Comment 1:

Can you please reconsider the title with a unique word as it seems so simple and less attractive than the audience.

Response:

Thank you for your valuable suggestion. We agree that a more specific and engaging title will help attract readers’ attention. We have revised the title to better emphasize the mechanistic focus and novelty of our findings.

Modification:

Before (Title-L1): ROS production and loss of ATP synthase subunit mRNA integrity as a mechanism of artificial deteriorated cotton seeds.

After: ROS-induced ATP synthase mRNA degradation and metabolism dysfunction reveals the mechanism of artificial deteriorated cotton seeds

Comment 2:

Abstract and conclusion are too simple; no novelty or significance is highlighted.

Response:

We appreciate the reviewer’s advice. We have rewritten the abstract and conclusion to highlight the novelty, emphasizing the mechanistic link between ROS accumulation and ATP synthase mRNA integrity loss, and its implications for seed vigor and germplasm preservation.

Comment 3:

Please double-check grammar; overall language needs verification.

Response:

We have carefully revised the entire manuscript for grammar, tense consistency, and scientific style.

Comment 4:

Please rewrite the introduction or add constructive information to make it attractive.

Response:

We have enriched the Introduction by emphasizing the gap in understanding mRNA oxidation during cotton seed aging.

Comment 5:

Add graphical/pictorial representation of your work along the abstract.

Response:

We have prepared a Graphical Abstract illustrating the sequence: ROS accumulation → mRNA oxidative damage → impaired ATP synthesis → Mitochondrial dysfunction. The figure provides an intuitive summary of our findings.

(A figure are provided below this document.)

Comment 6:

Graphs are too small and fonts unclear.

Response:

All figures have been resized to improve readability(Results section, Figs. 1–10).

Comment 7:

Add more justification in discussion and include relevant new findings.

Response:

We have expanded the Discussion to better connect ROS accumulation, mitochondrial dysfunction, and ATP synthase mRNA damage. We also referenced recent literature supporting the role of mRNA oxidation in post-transcriptional regulation during seed aging.

Comment 8:

Ensure references follow journal pattern.

Response:

All references have been reformatted according to PLOS ONE guidelines, including DOI addition and consistent author-year style.

Response to Reviewer 2

Comment 1:

Explain why cultivar “Xinluzao 74” was chosen.

Response:

We have clarified this in Introduction “Xinluzao 74” is a major high-yield, disease-resistant cultivar widely cultivated in Xinjiang, representing typical seed storage behavior of upland cotton. Its characteristics make it an ideal model for artificial aging studies with global relevance.

Comment 2:

Justify 45°C and 100% RH conditions.

Response:

We have added in Method part:

“These conditions follow Tesnier et al. (2002) and are widely used to simulate accelerated seed aging, providing reproducible deterioration kinetics comparable across oilseed species.”

Comment 3:

Expand figure legends for stand-alone clarity.

Response:

Revised all figure legends (Figs. 1–10) to include species, treatment, replicates, and statistical test information, ensuring they can be understood independently.

Comment 4:

Quantify TTC staining intensity.

Response:

We appreciate the reviewer's suggestion to quantify the TTC staining intensity. While we agree that quantification can be valuable, the visual difference in staining between the experimental and control groups is consistently clear and striking across all replicates, as shown in Figure 2. Furthermore, we would like to highlight that the dehydrogenase activity assay (presented in Figure 3 provides a precise and quantitative measurement of the same biological process that the TTC staining visually represents. We believe that this enzymatic data robustly supports our qualitative TTC observations.

Comment 5:

Discuss conductivity—possible non-aging effects.

Response:

While conductivity increase mainly reflects membrane leakage from aging, minor variation due to seed coat permeability cannot be excluded and warrants further study, so we have added clarification in case any possible misunderstanding.

Comment 6:

Interpret initial H₂O₂ decline more carefully.

Response:

The transient drop in H₂O₂ after 5 days might also result from peroxidase-mediated conversion or ROS-induced degradation, rather than exclusive scavenging, so we have added clarification in the Discussion part.

Comment 7:

Clarify whether ATP synthase and respiration decline are cause or consequence of mRNA damage.

Response:

The observed reduction in ATP synthase activity is likely a downstream effect of mRNA integrity loss, suggesting that ROS-induced mRNA oxidation may precede metabolic inhibition, so we added mechanistic clarification in Discussion part:

Comment 8:

Improve discussion—avoid repetition.

Response:

We revised the first two Discussion paragraphs to merge redundant result descriptions and focus on mechanistic interpretation and broader biological implications.

Comment 9:

Minor English issues.

Response:

Thanks for your comments, we have rephrased all informal expressions.

---

## [Decision Letter · Decision Letter 1]

16 Dec 2025

ROS-induced ATP synthase mRNA degradation and metabolism dysfunction reveals the mechanism of artificial deteriorated cotton seeds

PONE-D-25-49082R1

Dear Dr. Chen,

We’re pleased to inform you that your manuscript has been judged scientifically suitable for publication and will be formally accepted for publication once it meets all outstanding technical requirements.

Kind regards,

Smita Kumar, Ph.D.

Academic Editor

PLOS One

Additional Editor Comments (optional):

Reviewers' comments:

Reviewer's Responses to Questions

**Comments to the Author**

Reviewer #1: All comments have been addressed

Reviewer #2: All comments have been addressed

2. Is the manuscript technically sound, and do the data support the conclusions?

Reviewer #1: Yes

Reviewer #2: Yes

3. Has the statistical analysis been performed appropriately and rigorously?

Reviewer #1: Yes

Reviewer #2: Yes

4. Have the authors made all data underlying the findings in their manuscript fully available?

Reviewer #1: Yes

Reviewer #2: Yes

5. Is the manuscript presented in an intelligible fashion and written in standard English?

Reviewer #1: Yes

Reviewer #2: Yes

Reviewer #1: Response to Reviewer 1

Comment 1:

Can you please reconsider the title with a unique word as it seems so simple and less

attractive than the audience.

Response:

Thank you for your valuable suggestion. We agree that a more specific and engaging

title will help attract readers’ attention. We have revised the title to better emphasize

the mechanistic focus and novelty of our findings.

Modification:

Before (Title-L1): ROS production and loss of ATP synthase subunit mRNA integrity as

a mechanism of artificial deteriorated cotton seeds.

After: ROS-induced ATP synthase mRNA degradation and metabolism dysfunction

reveals the mechanism of artificial deteriorated cotton seeds

Comment 2:

Abstract and conclusion are too simple; no novelty or significance is highlighted.

Response:

We appreciate the reviewer’s advice. We have rewritten the abstract and conclusion to

highlight the novelty, emphasizing the mechanistic link between ROS accumulation

and ATP synthase mRNA integrity loss, and its implications for seed vigor and

germplasm preservation.

Comment 3:

Please double-check grammar; overall language needs verification.

Response:

We have carefully revised the entire manuscript for grammar, tense consistency, and

scientific style.

Comment 4:

Please rewrite the introduction or add constructive information to make it attractive.

Response:

We have enriched the Introduction by emphasizing the gap in understanding mRNA

oxidation during cotton seed aging.

Comment 5:

Add graphical/pictorial representation of your work along the abstract.

Response:

We have prepared a Graphical Abstract illustrating the sequence: ROS accumulation →

mRNA oxidative damage → impaired ATP synthesis → Mitochondrial dysfunction. The

figure provides an intuitive summary of our findings.

(A figure are provided below this document.)

Comment 6:

Graphs are too small and fonts unclear.

Response:

All figures have been resized to improve readability(Results section, Figs. 1–10).

Comment 7:

Add more justification in discussion and include relevant new findings.

Response:

We have expanded the Discussion to better connect ROS accumulation, mitochondrial

dysfunction, and ATP synthase mRNA damage. We also referenced recent literature

supporting the role of mRNA oxidation in post-transcriptional regulation during seed

aging.

Comment 8:

Ensure references follow journal pattern.

Response:

All references have been reformatted according to PLOS ONE guidelines, including

DOI addition and consistent author-year style

ALL COMMENTS HAVE BEEN PROPERLY ADDRESSED IN CURRENT MANUSCRIPT

GOOD LUCK

Reviewer #2: The authors have significantly improved the manuscript and addressed all comments, the paper should be accepted.

**Do you want your identity to be public for this peer review?** For information about this choice, including consent withdrawal, please see our Privacy Policy

Reviewer #1: **Yes:** Dr. Beenish Afzal

Reviewer #2: **Yes:** Muhammad Amjad Ali

---

## [Editor Report · Acceptance letter]

PONE-D-25-49082R1

PLOS One

Dear Dr. Chen,

I'm pleased to inform you that your manuscript has been deemed suitable for publication in PLOS One. Congratulations! Your manuscript is now being handed over to our production team.

Kind regards,

on behalf of

Dr. Smita Kumar

Academic Editor

PLOS One